# Roles of Heart Rate Variability in Assessing Autonomic Nervous System in Functional Gastrointestinal Disorders: A Systematic Review

**DOI:** 10.3390/diagnostics13020293

**Published:** 2023-01-12

**Authors:** M. Khawar Ali, Jiande D. Z. Chen

**Affiliations:** Division of Gastroenterology and Hepatology, University of Michigan School of Medicine, Ann Arbor, MI 48109, USA

**Keywords:** autonomic nervous system, heart rate variability, gastroesophageal reflux disease, gastroparesis, functional dyspepsia, irritable bowel syndrome, constipation

## Abstract

Functional gastrointestinal disorders (FGID) and gastroesophageal reflux (GERD) disease affect a large global population and incur substantial health care costs. Impairment in gut-brain communication is one of the main causes of these disorders. The central nervous system (CNS) provides its inputs to the enteric nervous system (ENS) by modulating the autonomic nervous system (ANS) to control the gastrointestinal functions. Therefore, GERD and FGID’s might be associated with autonomic dysfunction, which can be identified via heart rate variability (HRV). FGIDs may be treated by restoring the autonomic dysfunction via neuromodulation. This article reviews the roles of HRV in the assessment of autonomic function and dysfunction in (i) gastroesophageal reflux (GERD), and the following FGIDs: (ii) functional dyspepsia (FD) and gastroparesis, (iii) irritable bowel syndrome (IBS) and (iv) constipation. The roles of HRV in the assessment of autonomic responses to various interventions were also reviewed. We used PUBMED, Web of Science, Elsevier/Science direct and Scopus to search the eligible studies for each disorder, which also included the keyword ‘heart rate variability’. The retrieved studies were screened and filtered to identify the most suitable studies using HRV parameters to associate the autonomic function with any of the above disorders. Studies involving both human and animal models were included. Based on analyses of HRV, GERD as well as the FGIDs were found to be associated with decreased parasympathetic activity and increased sympathetic nervous system activity with the autonomic balance shifted towards the sympathetic nervous system. In addition, the HRV methods were also reported to be able to assess the autonomic responses to various interventions (mostly neuromodulation), typically the enhancement of parasympathetic activity. In summary, GERD and FGIDs are associated with impaired autonomic dysfunction, mainly due to suppressed vagal and overactive sympathetic tone, which can be assessed noninvasively using HRV.

## 1. Introduction

GERD and FGIDs are chronic disorders of the gastrointestinal (GI) tract, attributed to motility disturbances, visceral hypersensitivity, distorted mucosal and immune function, altered gut microbiota, and/or impaired central nervous system processing of the sensory input [1]. These disorders arise mainly due to impaired gut-brain communication without any identifiable biomarkers or structural abnormalities of the GI tract; therefore, their identification and classification are based on the associated symptoms [2,3]. In 1993, a study [4] revealed that 69% of US adults have at least one FGID. According to a recent study in 2021, including 73,076 individuals in 33 countries in four different continents, FGIDs have a worldwide prevalence of 40.3%, ranging from low of 33.7% in Singapore to 50% in Egypt, and they are more common in females than males. Of all of the participants included in the survey, around 46.5% of females and 34.3% of males were suffering from at least one FGID. Subgroup analysis revealed that the most common type of FGID is constipation, with its prevalence rate of 11.7%, followed by functional dyspepsia (7.2%), proctalgia fugax (5.9%), functional diarrhea (4.7%), and IBS (4.1%) in all of the regions included in the survey. Due to their impact on the quality of life (QOL) of the patients, FGIDs pose a substantial economic burden on the healthcare system worldwide [5,6,7].

The ROME diagnostic criteria is a widely acceptable method for FGID’s classification. It divides FGIDs into 33 adult and 20 pediatric disorders [5,8]. The most common types of FGID are: functional dyspepsia (FD) and gastroparesis, irritable bowel syndrome (IBS) and constipation. GERD, which is one of the most common GI disorders with a high prevalence (20%) in western countries, refers to the physiologic passage of the gastric content into the esophagus associated with heart burn or acid regurgitation. Other symptoms of GERD include hoarseness, globus, chronic cough, and dental erosion. It is usually diagnosed by endoscopy and 24 h ambulatory pH monitoring [9,10,11]. Gastroparesis and functional dyspepsia have overlapping symptoms, and both are neuromuscular disorders of the stomach involving dysfunctions of either afferent or efferent nerves or both. They are associated with symptoms, such as nausea, vomiting, epigastric pain, bloating, postprandial fullness with decreased or impaired gastric motility, and reduced gastric accommodation, delayed gastric emptying, and gastric hypersensitivity [12,13,14]. IBS is characterized by abdominal pain or discomfort, combined with the impaired bowel function in the absence of detectable structural abnormalities, which makes it a disorder caused by impaired gut-brain communication [15,16,17]. Constipation, which is considered as the most prevalent functional gastrointestinal disorder, is defined as less than three bowel movements per week, with or without straining to defecate, lumpy or hard stool, the sensation of incomplete evacuation, and/or obstruction.

### 1.1. Gut-Brain Communication

The GI tract is innervated by the intrinsic enteric nervous system (ENS) and extrinsic autonomic nervous system (ANS) [18,19]. While the GI functions display a significant extent of automaticity through the ENS, the central nervous system (CNS) provides extrinsic neural inputs by interacting with the ENS via autonomic innervations to regulate, modulate, and control GI functions [20,21,22]. Impairment in either the parasympathetic nervous system (PNS) or sympathetic nervous system (SNS) or both may result in impaired gut-brain communication, which can lead to functional GI disorders [18,23]. The parasympathetic innervation of the GI tract originates either from the vagus nerve or from the sacral parasympathetic nerves. The upper GI tract (esophagus, stomach, small intestine, ascending colon) receives its parasympathetic innervation from the vagus nerve while the lower GI tract (transverse colon, descending colon, sigmoid colon, the striated muscles of the external anal canal) have the parasympathetic innervations originating from the preganglionic neurons within the lumbosacral spinal cord, prominently from the S1–S4 regions [2,3,24]. Both the efferent and afferent fibers of the vagus nerve innervating the GI tract pass through the nucleus tractus solitarius (NTS) which also contains the autonomic innervations to and from the other organ whose function is controlled by the CNS via ANS modulation, such as cardiovascular function [24,25]. Therefore, the modulation of the autonomic activity by the CNS in response to the sensory inputs from the GI tract also affects the cardiovascular function because of the crosstalk between the subnuclei in the NTS modulating the rhythm or variability of the heart [24]. In earlier studies, we have shown that colonic contractions starting from the most proximal end of the ascending colon can be generated by injecting bisacodyl in the rectum. The chemical receptors in the rectum are activated upon sensing the bisacodyl and communicate to the higher brain centers by sacral afferent fibers through spinal pathways via Barrington’s nucleus and NTS. The CNS modulates in response to this sensory information and communicates with ENS by modulating the ANS in order to generate colonic contractions. This modulation of the ANS is visible in the cardiac autonomic output, and we successfully studied the change in sympathetic and parasympathetic activity before, during, and after the major colonic events via heart rate variability [25,26]. Hence, heart rate variability (HRV) can be used as a tool to study the autonomic activity in response to the GI functions and to identify the impaired gut-brain communication due to autonomic dysfunction, causing the FGID’s, as well as feedback to study the effect of treatment procedures applied to treat FGID’s. Conversely, the cardiac dysregulations may not mediate or cause FGID’s; however, the HRV of the patients with cardiac dysregulations should not be used to study the ANS modulation in association with the GI functions due the fact that the ECG recordings of such patients may be highly affected by cardiac dysfunction [27]. It should be noted that the HRV analysis is a surrogate of the GI autonomic function and caution should be taken in interpreting HRV findings.

It Is hypothesized that the autonomic nervous system plays a key role in normal gastrointestinal function, and autonomic dysfunction may lead to impaired gut-brain communication, which is one of the major causes of functional GI disorders. In current clinical practices, ANS is rarely assessed in treating GI disorders, the FGIDs may be treated by restoring the normal autonomic function as a way to restore the impaired gut-brain communication. Neuromodulation by electric or light stimulations may prove to be successful therapies to restore the autonomic dysfunction and hence the treatment for FGID’s.

### 1.2. Assessment of the Autonomic Function

Traditionally, the ANS assessment has been conducted by recording the nerve activity of the sympathetic nerve via electrodes placed invasively on the nerve fibers. This method of ANS assessment involved complications, especially in humans. Several other methods have been developed for the assessment of the sympathetic and parasympathetic activity of the ANS, including baroreflex sensitivity, plasma catecholamines, cardiac sympathetic neuroimaging, electrodermal activity, and heart rate variability. It has been established that changes in the baroreflex function represents changes in the autonomic control of the cardiovascular system; several methods have been developed to study the baroreflex functions in humans; however, most of these methods have limited value in clinical settings [28]. Plasma catecholamine measure the epinephrine, norepinephrine, and dopamine in the blood to record the neural activity of the autonomic nervous system [29]. Cardiac sympathetic neuroimaging uses the injection of compounds to radiolabel the catecholamine storage sites by tomographic radionuclide imaging [30]. Electrodermal activity records the autonomic activity from skin conductance modulation, which is induced by sweat gland activity [31]. With pros and cons in every method, heart rate variability is widely accepted as a powerful tool in autonomic assessment due to its accuracy, reproducibility, and simplicity [32,33]. As the ANS modulation to control the GI function has successfully been shown via heart rate variability in our earlier studies [25,26], it may not relate well in other methods of ANS assessment. Therefore, we compiled the studies using HRV to assess the autonomic function associated with FGID’s in this review.

### 1.3. Heart Rate Variability-Based Assessment of the Autonomic Function

Several time-domains, frequency-domains, and non-linear parameters of HRV have been developed and used to assess the sympathetic and parasympathetic nervous system activities using the RR interval signal generated from the electrocardiogram (ECG).

The time domain parameters quantitatively measure the amount of variability in heart rate using peak to peak or RR intervals data, which is the time duration between the successive R peaks of the ECG signal. These include the root mean square of successive differences of RR intervals (RMSSD), the percentage of successive RR intervals that differ by more than 50 ms (pNN50), standard deviations of: (i) the RR interval (SDRR), (ii) the normal RR intervals after removing artifacts (SDNN), (iii) the average NN intervals for each 5-min segment for a 24 h recording (SDANN), and its mean (SDNN index). RMSSD, pNN50, SDRR, and SDNN are directly correlated to parasympathetic activity, while the SDANN and SDNN index are known to be influenced by both the sympathetic and parasympathetic nervous system activity (Table 1) [34,35,36]. The pre-ejection period (PEP), which is the time duration between the electrical activity and mechanical activity of the heart, calculated from ECG and impedance signals (time duration between the Q point on the ECG signal and B point on the impedance cardiography signal), is a measure of SNS activity, but it is not a very sensitive parameter [26]. The sympathetic index (SI) is used as a measure of sympathetic nervous system activity; it is derived from the RR interval histogram [24,26,37,38]. A Poincare plot is a non-linear method to measure the autonomic activity; the length of the minor axis of the fitted eclipse (SD1) represents the PNS, while the length of the major axis (SD2) is influenced by both the SNS and PNS activity, while their ratio (SD2/SD1) represents the autonomic balance [25,26]. The ratios SI/RSA and SI/RMSSD have also been used to represent the autonomic balance [25,26,35]. All of these parameters are summarized in Table 1 below.

Frequency-domain parameters are derived from the spectral analysis of the peak-to-peak interval (RR interval) signal. The power spectral density of the RR interval signal is calculated using the fast Fourier transform and is divided into an ultra-low frequency band (ULF, ≤0.003 Hz), very low frequency band (VLF, 0.0033–0.04 Hz), low frequency band (LF band), and high frequency band (HF band). Figure 1 below shows the power spectral density of the RR interval signal with VLF, LF, and HF bands. The ULF band characteristics are not fully understood; the VLF band is known to be influenced by thermal and hormonal controls, along with vasomotor activity, and is not associated with the autonomic nervous system [36]. The parasympathetic activity is represented by high frequency (HF) power, which is the power of the frequency band ranging from 0.15–0.4 Hz for humans in the power spectrum of an HRV signal [26,35]. The HF is also represented as the HF peak power, HF power in normalized units (hFnu), HF percentage power, and natural log of HF power (respiratory sinus arrythmia or RSA). The LF power (0.04–0.15 Hz for humans) is known to be influenced by both the sympathetic and parasympathetic nervous system and the ratio of the LF power and HF power (LF/HF) is used to represent autonomic balance [26]. Both LF and HF bands are required to be adjusted based on the respiration frequency to include the respiration band in the HF power band in case of slow (<9 breaths/min) or very fast breathing (<24 breaths/min); the adjusted HF and LF bands are termed as high frequency area (hFa) and low frequency area (lFa) [39]. The definitions of the LF and HF should also be adjusted for different species. For example, in rats, LF is defined as 0.3–0.8 Hz and HF is defined as 0.8–4.0 Hz [40,41].

This systematic review is focused on the HRV-based assessment of the autonomic function in GERD and three common FGIDs, including FD and gastroparesis, IBS, and constipation.

## 2. Methods

Each disorder’s name was searched in combination with the keyword “Heart Rate Variability” in PubMed, Web of Science, Elsevier/Science Direct and Scopus. For example, in the case of constipation, the keywords “constipation” and “Heart Rate Variability” were searched. After retrieving the research articles, excluding the review articles, conference abstracts, and articles in languages other than English in each case, we shortlisted the studies. The shortlisting process included removing the studies whose focus was not on the searched disorder and/or on heart rate variability, for example, a study by Finsterer et al. was retrieved in the pool of constipation and HRV, but its focus was entirely different, [43]. In the next step, the studies which presented the results using at least one HRV parameter in association with the GI disorder under consideration or its treatment (i.e., neuromodulation) were included in this review. After finalizing the selection process, the details of each included study, such as study type (animal or human study), population (number of patients and/or healthy controls and their age and subgroups details if any), study protocol, HRV parameters used, the major HRV results, and the conclusion from those results, were collected from each selected study and tabulated in Table 2, Table 3, Table 4 and Table 5. The whole selection process and data collection process was conducted manually and independently by the first author and independently rechecked and verified by the second author. As mentioned above, this review is conducted for the three major FGIDs and GERD; the following procedure was applied to search, evaluate, and include the studies in each case.

### 2.1. Gastroesophageal Reflux

Although the gastroesophageal reflux disease (GERD) is not considered as an FGID according to the Rome IV criteria, it is included in this review because (1) clinically, GERD is most commonly determined based on symptoms; (2) the major pathophysiologies of GERD are functional, such as increased transient lower esophageal sphincter relaxation, low lower esophageal sphincter pressure, and impaired esophageal motility. These are similar to FGIDs. PUBMED, “Web of Science” and “Elsevier/Science Direct/Scopus” were searched with the keywords ‘gastroesophageal reflux’ and ‘heart rate variability’. In total, 29 studies were retrieved from PubMed with both keywords in their title and/or abstract, 14 from Web of Science, whose topic was heart rate variability and contained “gastroesophageal reflux disease” in their titles, 10 in Elsevier/Science Direct, and 10 in Scopus. Most of the studies were common in all searches and, after removing the duplicates, only the studies measuring the autonomic nervous system with at least one heart rate variability parameter comparing ANS activity of GERD patients with controls and/or among subgroups of patients or in association with some treatment of GERD were selected, which reduced the total number of included studies to 15.

### 2.2. Functional Dyspepsia and Gastroparesis

For these disorders, the keyword ‘heart rate variability’ was searched in combination with ‘Functional Dyspepsia’, as well as “Gastroparesis” in titles using PubMed, Web of Science, Elsevier/Science direct, and Scopus in the first step and 24 results were retrieved in PubMed, 4 in Elsevier, 6 in Web of Science, and 4 in Scopus. The review articles, conference abstracts, and papers in languages other than English were excluded. Most of the studies were common in all searches and, after removing the duplicates, only the studies measuring the autonomic nervous system with at least one heart rate variability parameter comparing the ANS activity of the functional dyspepsia or gastroparesis patients with healthy controls and/or among different subgroups of patients or in association with a treatment of functional dyspepsia or Gastroparesis were selected. The total number of included studies was 19.

### 2.3. Irritable Bowel Syndrome

A total of 49 results were searched by combining the keywords ‘Heart rate variability’ with ‘Irritable Bowel Syndrome’ in the title in PubMed, 20 in Web of Science, 17 in Elsevier, and 14 in Scopus. The review articles, conference abstracts, and papers in languages other than English were excluded. Most of the studies were common in all searches, and after removing the duplicates, only the studies measuring the autonomic nervous system with at least one heart rate variability parameter comparing ANS activity of the IBS patients with healthy controls and/or among different subgroups of patients or in association with a treatment of IBS were selected, which reduced the total number of included studies to 19.

### 2.4. Constipation

A total of 16 papers were searched by PUBMED with a combination of keywords ‘Constipation’ in the title and ‘Heart rate variability’ in both the title and abstract; 14 were retrieved in Scopus, 6 in Elsevier, and 14 in Web of Science. After removing the review articles, conference abstracts, and papers in languages other than English, the numbers reduced to 3 in Elsevier/Science direct, 13 in Scopus, 10 in Web of Science, and 16 in PubMed. Most of the studies were common in all searches and, after removing the duplicates, the studies measuring the autonomic nervous system with at least one heart rate variability parameter comparing ANS activity of the IBS patients with healthy controls and/or among different subgroups of patients or in association with a treatment of IBS were selected, which reduced the total number of included studies to 15.

After selecting the studies, each paper was analyzed for the population (or participants humans or animals) included in the study with their age, gender, and subgroups, if any. The design of each study and the heart rate variability parameters used to assess the autonomic nervous system activity were recorded in each case. Most of the studies compared the HRV parameters between healthy controls and groups of patients to assess the autonomic function or dysfunction in patients. While others compared the HRV parameters’ values before and after some intervention in animals or patients. The conclusion from each study was recorded based on the association/disassociation of autonomic function or dysfunction with the functional GI disorder under investigation. The data collection process was conducted manually and independently by the first author and reviewed and verified by the second author.

## 3. Results

### 3.1. Gastroesophageal Reflux and Autonomic Nervous System

A total of 861 individuals were included in all the 15 studies selected for the gastroesophageal reflux group. Seven out of the 15 studies compared the autonomic nervous system activity of the patients to that of healthy controls, while in the other seven studies, the comparison was carried out among different patient groups; three studies were conducted on newborn babies or kids while the rest were conducted on adults.

Four studies used short term HRV (≤5 min); seven studies used long term HRV (12–24 h); four studies did not mention the duration of the ECG recording. One study used only time domain parameters (SDNN, SDNNi, RMSSD, pNN50%) for the ANS activity assessment [50] and one study used heart ratios (max HR/Min HR) during postural change, Valsalva, hand grip, and deep breathing to record the variability in heart rate instead of time or frequency domain parameters [49]. Seven studies used only frequency domain parameters, HF and LF powers, and their ratios for the ANS assessment, and in the remaining six studies, both time and frequency parameters were implemented.

Based on the HRV parameter values, six studies concluded that patients had low overall heart rate variability compared to controls, as the HRV parameters (PNS and SNS parameters) were lowered in patients. Four studies reported a decreased parasympathetic activity and enhanced sympathetic tone with the autonomic balance shifted towards the SNS in patients compared to controls. One study compared two groups of patients and tried to differentiate alcoholic non cardiac chest pain patients with and without depression and alexithymia using HRV and could not find any difference [51]. Nowak et al. concluded that the children suffering from IBS had no significant differences in their autonomic activity compared to controls [53].

Djeddi et al. studied the autonomic activity before, during, and after the gastroesophageal reflux event and found that the PNS activity is significantly suppressed prior to the reflux event [9]. Jones et al. indicated that the autonomic neuropathy, which was believed to be the cause of gastroesophageal reflux, could be reversed if treated [49].

Three studies assessed the effect of esophageal electric stimulation on the autonomic nervous system in patients with GERD. Two of the three reported that transcutaneous electric acustimulation (stimulation delivered via acupoints) decreased the SNS and promoted the PNS activity in patients with GERD [11,52]. The other showed that electrical esophageal stimulation (EOS; 200 μs, 0.2 Hz., 25 stimuli) applied 5 cm above the lower esophageal sphincter altered ANS such that the sympathetic outflow (LF peak power) was decreased and the cardiovagal activity (HF peak power) was increased in patients with GERDS, but not in healthy controls [48]. This might be attributed to the fact that the baseline ANS activity in the healthy controls was already in a homeostatic state, hence the electric stimulations did not modulate it much, contrary to patients.

The detailed information on the population, study design, HRV parameters used, results, and conclusion in the 15 studies is summarized in Table 2.

### 3.2. Functional Dyspepsia, Gastroparesis and Autonomic Nervous System

Out of the total 19 studies included for the “functional dyspepsia and gastroparesis” group, six were animal studies, which included five studies on rats/rodents and one study carried out on dogs. In the 13 human studies, a total of 873 participants took part, which includes 154 healthy controls and 719 patients. Further details about the participants (gender, average age, pathophysiology of each group) are given in Table 3 below.

Thirteen studies utilized frequency domain parameters (HF power, LF power, LF/HF ratio, HFa, LFa, LFa/HFa) for autonomic assessment while five studies used both time (SDNN, SDANN, pNN50%, RMSSD, E/I, Valsalva ratio) and frequency domain parameters, and one study used only the time domain parameters for the ANS assessment.

While Kumar et al. did not find any abnormality in the autonomic nervous system function in patients with delayed gastric emptying [68], all other human studies have found at least one abnormal HRV parameter value in patients with functional dyspepsia or gastroparesis compared to healthy controls. Only one study reported that patients with FD had a high parasympathetic tone and a lowered LF/HF ratio and that symptoms were more severe in patients with impaired PNS activity [13]. All other studies demonstrated that both FD and gastroparesis were associated with a decreased vagal tone and/or an increased sympathetic activity. Three human studies have shown that the suppressed vagal activity can be enhanced (increase in HF power) and SNS can be decreased by gastric electrical stimulation or gastric neuromodulation [67], transcutaneous auricular vagus nerve stimulation [59], and epalrestat (an aldose reductase inhibitor, used for treating diabetic neuropathy) administration [62].

Rodent studies showed that delayed gastric emptying was associated with a low HF power and high LF/HF ratio (sympathovagal balance) compared to controls, which was improved with sacral nerve stimulations [14,58], electroacupuncture at ST36 with implanted electrodes [60,63], or auricular electroacupuncture [61]. Similar findings were reported in dogs with electroacupuncture at ST36 [64].

Table 3 summarizes the population, study design, HRV parameters used, results, and conclusions of all 16 studies in functional dyspepsia and gastroparesis.

### 3.3. Irritable Bowel Syndrome and Autonomic Nervous System

All 19 studies included for the “Irritable Bowel Syndrome” group in this review were carried out on human participants. There was a total of 2163 participants included in all of the studies, combined with 1494 patients suffering from irritable bowel syndrome and 669 healthy controls. Further details about the participants (gender, average age, subgroups) are given in Table 4.

Thirteen studies implemented the frequency domain parameters (HF power, LF power, LF/HF ratio, HFnu, LFnu, VLFnu), one study used peak HF and LF powers [74], two studies used the natural log of HF and LF power [16,79], and one study used the percentage HF and LF powers [17]. Two studies used both time and frequency domain parameters for the ANS assessment [81,86].

While eight studies demonstrated a high sympathetic tone (increased value of LF power and/or LF/HF ratio) and low parasympathetic tone (lower value of HF power) in patients with irritable bowel syndrome compared to adult healthy controls, Jarrett et al. reported no difference in HRV parameters in children with functional abdominal pain (FAP) related to IBS compared to healthy children [78]. Orr et al. showed that the difference in the autonomic activity of IBS patients is only abnormal during rapid eye sleep and during walking [87]. Kano et al. and Adeyemi et al. reported opposite results, showing a low ln(LF/HF) ratio and a higher ln(HF power) in patients compared to controls [16,88]. However, an inflatable bag was placed inside the rectum during the experiment of Kano et al. which was inflated during the experiment, which might have caused the opposite autonomic response. Jarrett et al. tried to use HRV parameters to differentiate the IBS patients with diarrhea (IBS-D) and with constipation (IBS-C) [80] while Robert et al. compared the HRV of IBS patients with and without depressive symptoms [75]. Both of their results showed no difference in the HRV of between the compared groups. Jarrett et al. reported that the IBS patients with high sympathetic tone were less benefitted from cognitive therapies [76] while, according to Jung et al., HRV parameters could be used to study the effect of cognitive therapies for IBS [15]. Cai et al. reported that only the IBS patient with severe gut pain had high sympathetic tone and low parasympathetic tone [85]. Tanaka et al. established a strong correlation between adrenaline and HRV upon corticotropin release in healthy controls, which was absent in IBS patients [83]. Similarly, Pellissier et al. reported that healthy controls could be differentiated from the IBS patients based on the fact that, in controls, a higher HFnu value correlated with a lower evening salivary cortisol level, while this correlation was absent in the patients [82].

### 3.4. Constipation and Autonomic Nervous System

Of the 15 studies included for the “Constipation” group, three were animal studies which included two studies in rats [89,91] and one study in dogs [93]. The 12 human studies consisted of 197 healthy controls and 634 patients with constipation. Further details about the participants (gender, average age, pathophysiology of each group) are given in Table 5.

Ten studies only implemented the frequency domain parameters (HF power, LF power, LF/HF ratio, HFnu, LFnu) for the autonomic assessment, while five studies used both time (RMSSD, SDNN, pNN50%, sympathetic index) and frequency domain parameters (HF power, RSA, LF power, LF/HF ratio). It is noteworthy that three studies implemented the sympathetic index (SI) for sympathetic nervous system assessment instead of the LF power and two studies used SI/RSA or SI/RMSSD for recording autonomic balance [24,38,97].

The studies involving human participants can be divided into three subgroups. The first subgroup (n = 4) focused on identifying the autonomic function/dysfunction in constipated patient. Lijun et al., Miyagi et al., and J. Liu et al. have concluded that constipation is associated with high sympathetic tone (high SI) and low parasympathetic tone (decreased HF or RSA) and the autonomic balance shifted towards SNS activity (low LF/HF) [38,90,97]; they concluded that both SNS and PNS were impaired. Gondim et al. only reported a higher SNS tone in constipated patients [94]. Enevoldsen et al. tried to corelate GI transit time with HRV parameters and found no correlation [96].

Ding et al. tested biofeedback a method to improve autonomic nervous system activity in constipated patients, but they could not establish biofeedback as a way to treat constipation by modulating the ANS [99]. The remaining six human studies tested neuromodulation as a method to treat chronic constipation via modulating vagal and/or sympathetic activity. In an earlier study, neuromodulation of lumbar and sacral autonomic nerves by low level laser therapy was reported to modulate the autonomic nervous system such that PNS activity was enhanced and SNS activity was decreased in patients with GI disorders predominantly constipations [24]. Z. Liu et al. modulated the ANS via transcutaneous electric acustimulation (TEA) and successfully increased the HF power, decreased the LF power as well as LF/HF ratio [92]. While J. Liu et al. found that ANS was modulated with both actual and sham TEA when both combined with Macrogol 4000 Powder (MAC) and adaptive biofeedback, whereas MAC alone did not modulate the ANS [95]. Zhang et al., Wu et al., and Chen et al. have successfully modulated the ANS activity using neuromodulation as an effort to treat the chronic constipation [98,100,101]

Similarly in animal studies, Wang et al. demonstrated that Loperamide-induced constipation in rats could be treated by electroacupuncture (EA), which increased the HF power and decreased the LF/HF ratio [89]. Similar findings on autonomic modulation were reported with sacral nerve stimulation in the same rodent model of constipation [91]. Jin et al. found that impaired colonic motility induced by rectal distension in dogs was associated with a significant decrease in the HF power, which was reinstated by EA at ST36 [93].

## 4. Discussions

This systematic review has been conducted to determine the roles of HRV analyses in the assessment of autonomic function in patients with GERD or FGIDs and the autonomic responses to interventions used to treat these disorders. The majority of the reviewed studies revealed a decreased parasympathetic activity and/or an increased sympathetic activity assessed by HRV in patients with FGIDs and disorders of GI motility. In addition, HRV was also found to play an important role in the assessment of autonomic responses to various interventions (mostly neuromodulation therapies) used to treat FGIDs or GI dysmotility.

GERD and FGIDs are also called disorders of the brain-gut axis [102], and therefore autonomic dysfunction is common. However, there is a lack of a reliable method, especially a noninvasive method, for the assessment of the ANS. Accordingly, the noninvasive HRV analysis methods are attractive in assessing autonomic dysregulation in FGIDs and GERD. As summarized in this review, although various methods and parameters are used in the analysis of HRV, the majority of the studies reported consistent findings of reduced parasympathetic activity and sympathetic dominance in patients with FGIDs or disorders of GI motility. These findings are in agreement with the autonomic control of GI motility: the activation of parasympathetic activity enhances GI motility, whereas the activation of sympathetic activity inhibits GI motility.

In addition to assessing the ANS, the HRV analysis methods have also been found to be clinically useful in determining the ANS response to various therapeutic approaches used for FGIDs and GERD. Electrical neuromodulation has become an emerging therapy for these disorders [103]. Various studies have consistently shown that electrical neuromodulation improves GERD and FGIDs by improving the autonomic dysfunction assessed by HRV [11,45,64]. As shown in some recent studies, HRV parameters, such as HF, may be used as a biomarker in predicting the therapeutic effect of neuromodulation for FGIDs and GERD.

While HRV is attractive in assessing autonomic function and dysfunction, there are several disadvantages with the use of HRV. Firstly, there is a relatively large inter-subject variation in HRV analyses and there are no well-established normative HRV parameters in healthy controls [104]. A relatively large sample size is typically needed to differentiate a patient group from healthy controls, and it is difficult to differentiate among subgroups of a specific disease. This is one of major reasons accounting for controversial findings reported in the literature and also in this review, among other factors, such as the use of different HRV analysis methods and different parameter definitions, as well as heterogeneity of patient groups. Since the autonomic function is highly associated with the mood of a subject, HRV should be recorded in a well-controlled condition, such as the feeding status, recording environment, and possible surrounding interferences. Due to slow variations in the frequency of HRV, a relatively long duration, such as 30-min or longer, of the ECG recording should be acquired. On the other hand, however, a prolonged recording of the ECG, may be subjected to various compounding factors, such as a varying recording environment and the subject’s mood [105]. The other disadvantage with the use of HRV for assessing the autonomic function of the GI tract is that HRV reflects the cardiac autonomic function rather than GI autonomic function, although, in most cases, HRV can be used as a surrogate of GI autonomic function. Under some conditions, caution should be exercised while interpreting HRV findings. For example, the GI parasympathetic system is known to be activated upon food ingestion. However, the HRV analysis results in a reduced parasympathetic activity [106].

Although this review presents very important results, it has some limitations. The heterogeneity in the results was due to the fact that the included studies had different study designs, populations, and HRV parameters. The review included the studies focused on both diagnosis and treatment; therefore, a META analysis was not conducted. PRISMA criteria for reviews were not fully implemented; however, several requirements were fulfilled. Despite these limitations the overall conclusion remained almost the same in most of the studies.

In summary, the non-invasive HRV analysis methods (time-domain and/or frequency-domain) are reliable in assessing autonomic function/dysfunction in patients with FGIDs. They are especially useful in assessing the autonomic responses to various therapeutic interventions. However, caution should be taken in interpreting HRV findings, including inter-subject differences, analysis methods, and ECG recording conditions.

## Figures and Tables

**Figure 1 diagnostics-13-00293-f001:**
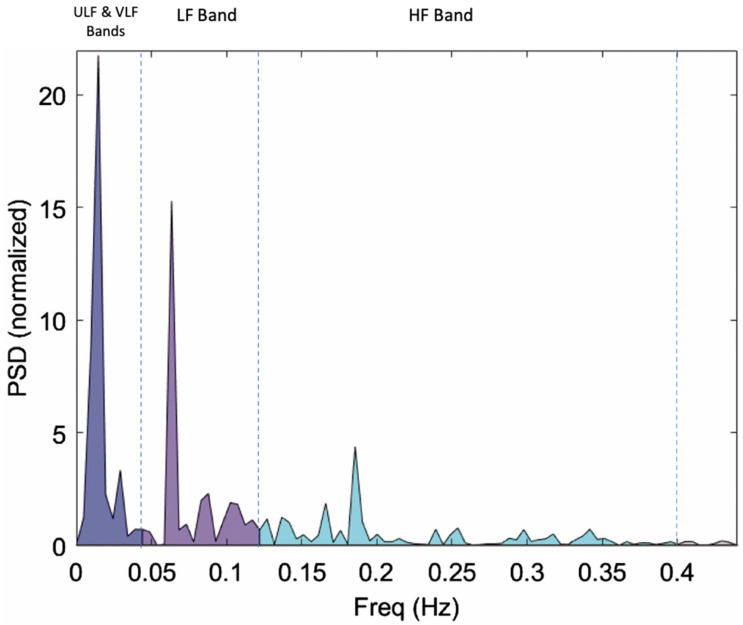
Power spectral density (PSD) of RR interval signal indicating ULF & VLF bands (≤0.04 Hz), LF Band (0.04–0.15 Hz), and HF Band (0.15–0.4 Hz) [Figure adapted from [42]].

**Table 1 diagnostics-13-00293-t001:** Heart rate variability (HRV) parameters.

HRV Parameter (Unit)	Description	Represents
RMSSD (ms)	Root mean square (rms) of the successive differences between the RR intervals $RMSSD=\sqrt{\frac{1}{n-1}\overset{n-1}{\underset{i=1}{\sum }}{\left(RR_{i+1}-RR_{i}\right)}^{2}}$ where *n* is the total number of *RR* intervals, *RR_i_* is the current *RR* interval, and *RR_i_*_+1_ is the next RR interval	Parasympathetic activity
pNN50 (*%*)	Percentage of successive RR intervals which differ by more than 50 ms	Parasympathetic activity
SDRR (ms)	Standard deviation of RR intervals	Parasympathetic activities
SDNN (ms)	Standard deviation of NN (RR intervals after removing artifacts from the recorded ECG signal)	Parasympathetic activities
SDANN (ms)	Standard deviation of averages of NN intervals in all of the 5 min segments of a 24-h recording	Influenced by both sympathetic and parasympathetic activities
SDNN index (ms)	Average of the standard deviations of all the NN intervals of each 5 min segment of a 24-h recording	Influenced by both sympathetic and parasympathetic activities
Pre-ejection period PEP (ms)	Time interval between the electrical activity (Q point on ECG signal) and mechanical activity (B point on the impedance signal) of the heart. PEP is inversely proportional to the sympathetic activity	Sympathetic activity
Sympathetic index SI (s^−2^)	Calculated from the histogram of the RR intervals using 50 ms bin size: $SI=\frac{AM_{o}*100\%}{2M_{o}*\;M_{x}DM_{n}}$ where *AM_o_* = amplitude of mode, *M_o_* = mode of RR intervals, *M_x_DM_n_* = range of RR intervals	Sympathetic activity
SD1 (ms)	The length of the minor axis of the fitted eclipse on the Poincare Plot of RR intervals $SD1=\sqrt{\frac{1}{N-1}\overset{N-1}{\underset{i=1}{\sum }}{\left\{\frac{RR_{i}-RR_{i+1}}{\sqrt{2}}\right\}}^{2}}$ *RR_i_* and *RR_i_*_+1_ represent the current and next RR intervals.	Parasympathetic activity
SD2 (ms)	The length of the major axis of the fitted eclipse on the Poincare Plot of RR intervals $SD2=\sqrt{\frac{1}{N-1}\overset{N-1}{\underset{i=1}{\sum }}{\left\{\frac{RR_{i}+RR_{i+1}-2\bar{RR}}{\sqrt{2}}\right\}}^{2}}$ *RR_i_* and *RR_i_*_+1_ represent the current and next RR interval, and $\bar{RR}$ is the mean value of RR intervals used	Influenced by both sympathetic and parasympathetic activities
SD2/SD1	The ratio of the lengths of major and minor axes of the Poincare plot of RR intervals	Autonomic balance
SI/RSA (s^−2^/ln(ms^2^))	Ratio of sympathetic index to respiratory sinus arrythmia	Autonomic balance
SI/RMSSD (s^−3^)	Ratio of sympathetic index to RMSSD	Autonomic balance

**Table 2 diagnostics-13-00293-t002:** Summary of Studies: Autonomic Nervous System and Gastroesophageal Reflux.

Sr.	Study	Population (n)	Study Design	HRV Parameters	Results	Conclusion
1	Lee et al., 2004 [44]	164 participants, including 57 controls, 34 with non-erosive reflux disease (NERD), 40 with symptomatic esophagitis (SE) and 33 with asymptomatic esophagitis (AE)	5-min resting autonomic activity was recorded via heart rate variability and compared between all of the patient and control groups.	SDNN, RMSSD, LF power, HF power, LF/HF power ratio.	HF power was significantly higher in patients with NERD in comparison with both SE and AE patient groups. All of the HRV parameters for autonomic tones were significantly lower in erosive as compared to non-erosive group. No difference was observed in time domain parameters.	The patients with endoscopically confirmed esophagitis (with or without symptoms) had low autonomic tone as compared to those with NERD.
2	J. Huang et al., 2013 [45]	17 patients with laryngopharyngeal reflux (LPR) and 19 healthy controls (age between 19 and 50)	5-min recording in supine position using blood volume pulse (BVP) sensor to record the heartbeat from the fingertip. 10 min adaption time before the actual recording.	RR interval, LF power, HF power, LF%, HF%, LF/HF	HF% was significantly lower (*p* = 0.003) and LF/HF ratio (*p* = 0.12) was significantly higher in patients with LPR, compared to healthy controls, indicating poor autonomic modulation and high sympathetic activity in patients. No significant difference in LF power and HF power in two groups.	Autonomic dysfunction seemed to be involved in the pathogenesis of LPR.
3	Djeddi et al., 2013 [9]	19 newborns with suspected gastroesophageal reflux (GER)	All participants underwent simultaneous 12-h polysomnography and esophageal pH monitoring. HRV parameters were recorded and compared during three types of periods (control, prior to and during the reflux)	RR, SDSD, RMSSD, pNN50, LF, HF, lFnu, hFnu, LF/HF	A significant increase in sympathovagal ratio (+32%, *p* = 0.013) was observed in the period prior to reflux which is caused by the 15% reduction in PNS activity (*p* = 0.017) relative to control period.	Gastroesophageal reflux events were preceded by a decrease in parasympathetic tone.
4	Milovanovic et al., 2015 [10]	29 patients (14 males and 15 females) aged (51.14 ± 18.34 years) with diagnosed GERD and 116 healthy controls	The study protocol included the evaluation of autonomic function and hemodynamic status, short-term heart rate variability (HRV) analysis, 24 h ambulatory ECG monitoring with long-term HRV analysis and 24 h ambulatory blood pressure monitoring.	*Short-term HRV* Average dRR, SD dRR, MD drr, pNN50%, RMSSD, VLF, LF, HF, LF/HF *Long-term HRV* Mean RR, SDNN, SDANN index, SDNN index, RMSSD, pNN50%, total power, VLF, LF, HF	*Short-term HRV analysis* All of the spectral and time domain HRV parameters were significantly lower in patients with GERD compared to controls, except for the LF/HF ratio. *Long-term HRV analysis* All of the HRV parameters had significantly lower values in patients with GERD compared to controls except for the mean RR.	GERD patients exhibited distortions of both the sympathetic and parasympathetic nervous systems with parasympathetic function appeared more congruent to GERD
5	Chen & Orr et al., 2004 [46]	12 GERD patients (6 males, 6 females; mean age = 37.2 ± 2.7 yrs) and 12 healthy controls (5 males, 7 females; mean age = 32.8 ± 2.2 yrs)	All the participants had 20-min water infusion (6 mL/min) and 20-min acid infusion (0.1 N HCL, 6 mL/min). ECG was recorded during each infusion period and HRV parameters were compared between the patients with GERD and control groups.	HF power, LF power, and LF/HF ratio (LF was considered as a measure of sympathetic activity)	Comparison of patient and control groups indicated that there was no significant change in LF band power in any period. The HF band power was significantly lower in GERD patients during all infusion periods. The LF/HF ratio was significantly larger in GERD patients.	Enhanced sympathetic dominance to esophageal acid infusion in patients with GERD.
6	Lee et al., 2006 [47]	30 GERD patients (21 males, 9 females; age range 28–83 years) with at least three episodes per week of heartburn and acid regurgitation. The patients were divided into two groups: pathological reflux group (n = 15) and functional heartburn group (n = 15)	All the participants referred for 48 h ambulatory pH monitoring underwent simultaneous 24 h cardiac monitoring for HRV to study the relationship between esophageal acid exposure and HRV.	LF power, HF power, LF/HF ratio	Patients with pathological reflux had lower average LF and HF powers than patients with functional heartburn. A significantly higher HF power and lower LF/HF ratio was found during sleep time, regardless of the diagnosis. The esophageal pH was positively associated with change in both LF and HF powers during waking and only HF power during sleeping. This association decreases with time.	Esophageal reflux was found to be associated with a decreased autonomic tone. A predominant parasympathetic fluctuation during sleeping and a superimposed sympathetic interaction during waking dictate during the daytime.
7	Yu et al., 2019 [11]	21 patients with refectory gastroesophageal reflux (rGERD) were divided into three groups with n = 7 in each group.	Group A received esomeprazole (ESO, 20 mg bid); group B received transcutaneous electrical acustimulation (TEA)+ deep breathing training (DBT) + ESO; group C received sham-TEA +DBT + ESO in a four-week study. ECG was recorded and HRV was evaluated at baseline and at the end of each treatment. Acetylcholine (Ach) and nitric oxide (NO) were measured from blood samples. Esophageal manometry and 24 h pH monitoring was performed before and after the treatment.	LF/(LF + HF) HF/(LF + HF)	Low frequency band (LF)/(LF + HF) ratio in groups B and C was decreased, compared with group A (*p* = 0.010, *p* = 0.042, respectively); high frequency band (HF)/(LF + HF) ratio in B and C groups was significantly increased, compared with group A (*p* = 0.010, *p* = 0.042, respectively). The serum Ach in groups B and C was significantly higher than group A (*p* = 0.022, *p* = 0.046, respectively); the serum NO in groups B and C was significantly lower than group A (*p* = 0.010, *p* = 0.027, respectively).	TEA combined with DBT significantly increased lower esophageal sphincter pressure (LESP), reduced acid reflux, and improved clinical indices of GERD. Both TEA and DBT significantly enhanced vagal activity and suppressed sympathetic activity assessed by the spectral analysis of HRV.
8	Hollerbach et al., 2000 [48]	12 healthy volunteers (1 female, 11 males aged 32 ± 8 yrs); 8 patients with non-cardiac chest pain (NCCP) (3 females, 5 males, aged = 40.5 ± 10 yrs)	Electrical oesophageal stimulation (EOS; 200 ls, 0.2 Hz, 25 stimuli) was applied to the oesophageal wall 5 cm above the lower oesophageal sphincter (LOS), and perception thresholds (measured in mA) were determined. EP responses were recorded using 22 standard electroencephalograms scalp electrodes. Heart Rate variability was used to assess the autonomic activity before, during, and after the oesophageal stimulation.	HF power, LF power, and LF/HF ratio (LF was considered as a measure of sympathetic activity)	Electrical stimulation decreased the sympathetic outflow and increased the vagal activity as represented by significantly higher values of LF area and HF area in patients compared to the healthy controls. LF/HF ratio decreased significantly during the stimulations in patients. The HR decreased in patients during EOS and not in controls.	NCCP patients in comparison with healthy controls modulated their autonomic nervous system in response to the oesophageal stimulations.
9	Jones et al., 2016 [49]	Control group included 71 participants (18M, age = 48.6 ± 11.9 yrs) 14 patients (8M, age = 49.1 ± 13.6 yrs) with non-erosive reflux disease (NERD) and 10 patients (6M, age = 51.6 ± 11.3 yrs) with erosive reflux disease (ERD)	HRV was assessed for all of the participants before starting a course of proton pump inhibitor (PPI) therapy and 8 weeks from the start of PPI therapy. The values were compared between patients and control groups.	*I/E Diff (bpm)* HR difference between maximum and minimum heart rate at 5 breaths/min *Valsalva ratio* *Hand grip ratio* Ratio of HR during maximal handgrip to heart rate during resting *Lying/Standing ratio* Ratio of maximum to minimum HR following standing *Deep Breathing* Analysis of HRV during 2 min of metronome-guided breathing at 6 breaths per minute.	ERD group had low HRV which improved significantly after PPI therapy as measured by I/E ratio on forced breathing, Valsalva ratio, and breathing at 6 breaths per minute. NERD group also shown increase in HRV after PPI but did not reach statistical significance.	The cardiac autonomic neuropathy, as measured by HRV, was associated with gastroesophageal reflux disease and th successful treatment of the inflammation could lead to reversal of the deterioration of autonomic tone.
10	Tirosh et al., 2010 [50]	17 infants diagnosed with idiopathic apparent life-threatening event (IALTE) and gastroesophageal reflux (GER) and 17 infants with only IALTE without GER were selected for this study. The age of all of the infants ranged from 3–28 weeks.	All of the participants Underwent a polysomnography, including esophageal pH measurements. Obstructive apneas with and without associated GER were identified. HRV was assessed using time domain analysis for short and long-term variability. Forty R-R intervals for each epoch before, during, and after the episodes, as well as 10 segments of 40 R-R intervals unrelated to apneic episodes were analyzed.	SDNN, SDNNi, RMSSD, pNN50%	Infants with IALTE and GER had low baseline short-term variability compared to the comparison group. All the HRV parameters (both short and long term) increased before, as well after the obstructive apnea as compared to baseline values. This was not observed in infants with both IALTE and GER.	Infants suffering from GER and IALTE had significant abnormalities in their autonomic control, as marked in coupled events of apnea and GER.
11	Swiatkowski et al., 2004 [51]	52 alcohol dependent males with chest pain *Subgroups:* with (n = 37, age = 40.9 ± 7.8 yrs) and without (n = 15, age = 41.9 ± 8.9 yrs) depression and alexithymia	Gastroduodenoscopy, esophageal and gastric pH, 24-h esophageal manometry, treadmill stress test, Holter monitoring, and blood sampling and HRV were recorded and compared between the two groups.	Mean HR, HR range, mean RR, SDNN, SDNNi, SDANNi, RMSSD, PNN50%, total power of HRV spectrum (TPS), LF, HF, VLF, LF/HF ratio.	No significant difference was observed in any HRV parameter in two groups.	Alcoholic patients with depression and alexithymia were more prone to functional disturbances in the upper GI tract. The differences between groups were not related to changes in ANS activity.
12	Hu et al., 2020 [52]	30 Patients with GERD	All the patients underwent two randomized sessions of TEA and sham-TEA at PC6 and ST36 acupoints with simultaneous esophageal high-resolution manometry (HRM), gastric accommodation, and ECG recordings and postprandial dyspeptic symptoms	HF power	HF power increased significantly during TEA in postprandial state. HF power was positively correlated with the percentage of normal slow waves and negatively correlated with regurgitation score.	TEA increased gastric accommodation and slow waves while decreasing postprandial fullness via mediating the vagal activity.
13	Nowak et al., 2017 [53]	16 children (aged 6–18 years), including 8 with asthma, 2 with GERD	24-h esophageal multichannel intraluminal impedance-pH and ECG were monitored simultaneously. Parasympathetic activity was calculated before, during and after the gastroesophageal reflux (GER) episodes	HF power, RMSSD	No change in HF power as well as RMSSD was observed before, during, and after the GER episodes	GER episodes were not associated with short term HRV or parasympathetic activity in children.
14	Floria et al., 2017 [54]	135 GI patients were divided into two groups: group I (n = 61 with GERD; age = 61.5 ± 9 yrs, 41% male, BMI = 28.8 ± 4 kg/m^2^) and group II (n = 74 without GERD; age = 58 ± 9 yrs, 46% male, BMI = 29 ± 4 kg/m^2^)	All patients underwent gastrointestinal endoscopy and 24-h ECG Holter monitoring.	SDNN, LF/HF ratio	SDNN was significantly lower in GERD group. LF/HF ratio was lower in GERD group without statistical significance.	Sympathovagal balance was disrupted in patients with GERD with PNS, contributing more to this disruption.
15	Campo et al., 2001 [55]	C group (12 healthy subjects); 28 patients with GERD (age = 41 yrs, range = 20–62 yrs) *Subgroups:* R group (GERD patients with dyspepsia; n = 21) NR group (n = 7; with normal pH results)	All participants underwent esophageal manometry, ambulatory 24-h pH study, and ECG recording.	LF/HF ratio	No difference in LF/HF ratio was observed in supine and upright position in both R and NR groups compared to the C group.	The sympathetic nervous system was lowered in patients with GERD based on the Blood pressure change during hand grip test. However, HRV did not differ from controls.

**Table 3 diagnostics-13-00293-t003:** Summary of Studies: Autonomic Nervous System and Functional Dyspepsia and Gastroparesis.

Sr.	Study	Population (n)	Study Design	HRV Parameters	Results	Conclusion
1	Guo et al., 2018 [56]	85 patients with functional dyspepsia (FD) with (age = 39.4 ± 12.82 years) and without (age = 36.5 ± 11.61 years) delayed gastric emptying.	HRV was recorded for 30 min before and after a meal. HRV parameters were compared for patients with and without delayed overall and proximal gastric emptying.	LF power, HF power, LF/HF	HF power was significantly lower and LF/HF ratio was significantly higher in patients with delayed proximal gastric emptying compared to those with normal proximal gastric emptying (GE) after the meal	Delayed GE was associated with disrupted sympathovagal balance due to decreased vagal activity after the meal.
2	I.-S Lee et al., 2018 [57]	15 patients (41 ± 4.72 years) with FD and 17 healthy controls (37.65 ± 4.02 years)	The physiological (including autonomic), emotional, and attentional response of FD patients and controls to high fat food, low fat food, and non-food images after taking an ad-libitum breakfast were compared.	SDNN, HF power, LF/HF ratio	FD patients showed higher SDNN value and lower LF/HF ratio (*p* < 0.05) compared to HC group. HF power showed no difference.	FD was associated with lower LF/HF ratio
3	Winston & Sarna, 2016 [1]	*Animal Study:* 12 rats with gastric hypersensitivity (GHS) and 12 control rats	The HRV of the rats was recorded for 30 min using BIOPAC system via a pair of electrodes placed under the skin overlaying the chest and behind the nape of neck.	LF/HF ratio	The GHS group had a significantly higher value of LF/HF ratio compared to the control group.	Neonatal programming triggered by colon inflammation increased the ratio of sympathetic to vagal tone (LF/HF)
4	Ye et al., 2020 [58]	*Animal Study:* 8 male Sprague-Dawley rats (450–500 g) with a chronically implanted gastric cannula and ECG electrodes	Beside testing the sham and sacral nerve stimulation (SNS) parameters and involvement of spinal efferent pathways via detecting c-fos immunoreactive cells in nucleus tractus solitarius (NTS), the involvement of vagal efferent activity was studied using spectral analysis of HRV signal.	HF power, LF/HF ratio	SNS at 5 Hz. increased the HF power and decreased sympathovagal balance (LF/HF ratio) compared to sham stimulations.	SNS with certain parameters improved gastric accommodation mediated via a spinal afferent and vagal efferent pathway.
5	Zhu et al., 2021 [59]	36 patients with FD (21F) and 39 healthy controls (23F; mean age = 43.8 yrs)	Patients were randomized into 2 groups (n = 18 each) to receive 2-wk taVNS or sham-ES. The dyspeptic symptom scales, anxiety and depression scores, and the same physiological measurements were assessed at the beginning and the end of the 2-wk treatment.	HF power, LF Power	Acute taVNS, significantly increased vagal activity (HF power) and decreased LF power during the 30-min postprandial period as compared to the sham-ES group. The HF was significantly increased (0.29 ±0.03 during taVNS vs. 0.24 ± 0.02 during sham-ES, *p* = 0.026) and the LF was significantly reduced (0.71 ± 0.03 during taVNS vs. 0.76 ± 0.02, *p* = 0.026) after acute taVNS in the 30-min postprandial period.	Compared with the HC, the patients showed decreased vagal activity. The non-invasive taVNS improved FD symptoms by improving gastric accommodation and gastric pace-making activity, as well as increasing vagal activity.
6	Zhang et al., 2020 [60]	*Animal Study:* 8 control rats and 8 FD Sprague-Dawley male rats. Stress was induced by placing the rats in cylindrical tube for 30 min.	8 weeks after treating the neonatal staged rats with intragastric iodoacetamide, the electrodes were implanted in these rats for the measuring gastric slow waves (GSW) and electrodes into acupoints ST36 for electroacupuncture (EA). The involvement of central afferent pathways was studied via detecting c-fos immunoreactive cells in nucleus tractus solitarius (NTS). Autonomic functions were assessed by spectral analysis of heart rate variability.	HF (0.8 -4.0 Hz.) power, LF (0.3–0.8 Hz.)//HF ratio	Electroacupuncture (EA) significantly increased HF power and decreased LF/HF ratio in FD rats under stress compared to the sham group.	EA at ST36 improved GSW under stress in FD rats mediated via the central and autonomic pathways.
7	Tominaga et al., 2016 [13]	45 FD patients [epigastric pain syndrome: EPS type, n = 24, 53.1 ± 2.1 years of age; 10 male and postprandial syndrome: PDS type, n = 21; 52.1 ± 3.3 years of age; 10 males]. 9 healthy volunteers with no GI symptoms (33.5 ± 4.3 yrs; 6 male)	24-h HRV was examined: the basal level, responses after lunch, cold presser and mental arithmetic tests, and the efficacy of the autonomic drug (tofisopam) were included in analysis.	HF power, LF/HF ratio.	The HF power was higher and LF/HF ratio was decreased in patients. GI symptoms were more severe in patients with ANS disorder (*p* = 0.085). The abnormal HF response after lunch occurred in 38.2% of patients who had greater tendency of indigestion (*p* = 0.061). Delay in recovery to the baseline ANS level after the cold pressor and the mental arithmetic tests occurred in a few patients. Tofisopam partly improved autonomic nervous system dysfunction and abdominal pain/indigestion.	Imbalanced ANS function and susceptibility in recovery from external stimuli were observed in functional dyspepsia patients, which was associated with dyspeptic symptoms.
8	Zhou et al., 2017 [61]	*Animal Study:* 10 rodents gavaged with 0.2 mL of 0.1% iodoacetamide in 2% sucrose. Control group had 8 rodents who were given only 0.2 mL of 2% sucrose. Age at experiment time = 8 weeks	Auricular electro-acupuncture (AEA) at the stomach point with different parameters or sham-EA was performed on 8-week-old rats. The sensitivity to gastric distention was recorded under different conditions. Autonomic functions were assessed from the spectral analysis of HRV	HF power, LF/HF ratio	FD rats had lower HF power and higher LF/HF ratio. The HF was increased, and LF/HF ratio was decreased by AEA	AEA reduced gastric hypersensitivity in FD rats by improving sympathovagal balance.
9	Okamoto et al, 2003 [62]	15 diabetic patients (7M, mean age = 59.1 ± 7.8 yrs.) 15 healthy controls (8M, mean age = 56.3 ± 6.5 yrs.)	ECG was recorded before and after the administration of epalrestat (DM group). ANS was evaluated by spectral analysis of HRV.	HF power, LF power, LF/HF ratio	The values of HF power, LF power, and LF/HF ratio were significantly lower in the DM group before administering eparestat as compared to HC which improved after administering eparestat.	Both vagal and sympathetic tones were reduced in patients with DM.
10	Yin et al., 2010 [63]	*Animal Study:* Sixty-three Sprague-Dawley rats (male, 300–350 g) were randomly divided into five groups. One of the five groups, group E had 10 rats with cutaneous electrodes for recording the HRV	Five experiments were performed in five groups of streptozotocin (STZ)-induced diabetic rats to study the effects of electroacupuncture (EA) at ST-36 on gastric slow-wave dysrhythmia, delayed GE and intestinal transit, impaired gastric accommodation, and the mechanisms of EA involving the autonomic and opioidergic pathways. Here we discuss the HRV results only.	HF power, LF power, LF/HF ratio	EA increased the vagal activity (HF power) and decreased LF/HF power compared to baseline.	EA at ST-36 increased vagal activity, resulting in improvement in gastric dysrhythmia, delayed GE and intestinal transit, and accommodation in STZ-induced diabetic rats.
11	Ouyang et al., 2002 [64]	*Animal Study:* 7 healthy female hound dogs (13–20 kg) that had undergone preparation with duodenal cannulation and placement of gastric serosal electrodes during a prior laparotomy	Electroacupuncture (EA) was performed from 30 min before until 45 min after the meal. Gastric myoelectrical activity and electrocardiogram were recorded. Vagal activity assessed from the spectral analysis of HRV recorded before during and after the EA.	HF power, LF/HF ratio	EA significantly increased the HF power and accelerated gastric emptying and decreased the LF/HF ratio.	The increased vagal activity due to EA could be involved in accelerating the gastric emptying.
12	Cai et al., 2008 [65]	27 female patients (60 yrs; range = 40–79 yrs) with primary Sjogren’s Syndrome (pSS). 25 female healthy controls (60 yrs; range = 42–79 yrs)	Beat-to-beat heart rate and blood pressure data was recorded in response to five standard cardiovascular reflex tests i.e., supine to standing ΔSBP, supine to standing 30/15 ratio, isometric grip ΔMBP, Valsalva ratio, and breathing E/I ratio.	Mean RR, pNN50%, LF power, HF power, breathing E/I ratio, Valsalva ratio, blood pressure variability (SBP)	The pSS patient group had several mild autonomic disturbances compared to control group i.e., decreased HRV, decreased BP variability, and increased heart rate, especially in response to postural change. The severity of the gastroparesis is corelated with decrease in HRV.	There was mild autonomic dysfunction (low HRV) associated with pSS.
13	Nguyen et al., 2020 [66]	242 patients with chronic gastroparetic symptoms which included 45 diabatic (13M, age = 45 ± 11 yrs) and 179 idiopathic patients (16M, age = 42 ± 13 yrs).	Baseline or resting HRV, sympathetic challenge (Valsalva), parasympathetic challenge (deep breathing), parasympathetic response to Valsalva or standing and sympathetic response to standing were recorded.	Lfa, Rfa, lFa/rFa	89% of diabatic and 74% of idiopathic patients had low sympathetic response to Valsalva or standing. Patients with delayed gastric emptying are more likely to have paradoxical parasympathetic excessive during Valsalva or standing. Patients having severe symptoms have parasympathetic dysfunction compared to those with mild symptoms.	ANS dysfunction was common in gastroparesis patients. PNS dysfunction was associated with severe upper GI symptoms and delayed gastric emptying and SNS dysfunction was associated with milder symptoms.
14	Stocker et al., 2016 [67]	rp A: 39 patients (6M, mean age = 38 years) Grp B: 35 patients (7M, mean age = 37 years) Grp C: 5 patients (0M, mean age = 48.6 yrs)	All of the patients underwent gastric neuromodulation. The autonomic response in Grp A was studied by systemic autonomic testing. Grp B was studied by both systemic autonomic testing and heart rate variability. Grp C suffering from diabatic gastroparesis was included in a pilot study to test their baseline autonomic activity by both systemic and HRV methods.	HF power, LF power	Both methods of autonomic testing predicted the abnormal baseline autonomic activity and improvement after neuromodulation in grp A and B. Pilot study indicated that both methods agree.	Both direct autonomic measurement and indirect measurement (HRV) predicted autonomic dysfunctions during baseline in patients with gastroparesis.
15	Kumar et al., 2021 [68]	89 patients (62M, mean age = 50 yrs ranged 38–57 yrs) with chronic kidney disease (CKD), with or without delayed gastric emptying (GE)	Patients were evaluated for gastroparesis symptoms via questionnaire, nutritional status, autonomic function via HRV and GE.	Mean RR, SDNN, RMSSD, VLF power, LF power, HF power, TP, LF/HF ratio, lFnu, hFnu	22/89 patients had delayed GE and 8/89 had rapid GE. No association between delayed GE and gastroparesis symptoms and autonomic neuropathy. All the HRV parameters were not statistically different in delayed and non-delayed patients.	No association between delayed GE and autonomic neuropathy in non-diabatic CKD patients.
16	Softeland et al., 2014 [69]	20 patients (5M, age = 44.5 ± 9.6 yrs) with diabetes and symptoms of gastroparesis 16 healthy controls (5M, age = 44.8 ± 9.3 yrs)	All of the patients were evaluated for gastric emptying rate followed by rectal sensory assessment and heart rate variability assessment and compared to that of controls.	Mean RR, SDNN, SDANN, pNN50%, RMSSD	60% of patients had delayed gastric emptying. The rectal hypersensitivity was also prevalent in patients compared to controls. The HRV parameters were low in patients compared to that of controls. Shortened RR interval correlated with reduced rectal temperature sensitivity.	The patients with signs and symptoms of diabetic gastroparesis, rectal sensitivity had decreased HRV.
17	Varges-Luna et al., 2020 [70]	15 FD patients (age: 8–17 yrs) 12 healthy controls with matching age range	ECG was recorded for 30-min before and 60-min after a cold pressor task (CPT). Gastric ECG and HRV parameters were calculated before and after CPT and in short intervals.	SDNN, RMSSD, pNN50%, LF power, HF power LF/HF	No significant change in RMSSD, pNN50%, HF, LF, and LF/HF ratio was observed in pre and post CPT in both patients and controls. SDNN increased significantly from pre to post CPT in controls but not in FD patients.	CPT did not induce any change in HRV of FD patients in youth. Youth with FD lacked normal flexibility in ANS in response to physical stressor.
18	Friesen et al., 2010 [71]	*1st part:* Children with FD (n = 9) Healthy children (n = 28) *2nd part:* Children with FD (n = 8) Healthy children (n = 26)	In 1st part, the ECG was recorded and HRV parameters were calculated for all the participants for 30-min pre and 60-min post meal. In second part, the ECG was recorded and HRV parameters were calculated for all the participants for 30-min pre and 60-min post rapid water loading.	HF power, LF power, LF/HF ratio	HF power was high and LF power and LF/HF ratio were low in FD patients post meal. Baseline LF/HF was positively correlated with water load volume in controls and negatively correlated in FD patients.	FD was associated with abnormal autonomic response to meal and water loading in children.
19	Silva Lorena et al., 2002 [72]	23 FD patients (7F, mean age = 38 ± 7 yrs) and 12 healthy controls (7F, mean age = 34 ± 4 yrs)	24 h ECG was recorded and HRV parameters were calculated and compared between FD patients and controls.	LF power, LF%, lnLF, HF power, HF%, lnHF, LF/HF, SDNN, RMSSD, pNN50%	HF power, HF%, lnHF, and RMSSD were significantly lowered in FD patients compared to healthy controls.	FD patients had impaired vagal function.

**Table 4 diagnostics-13-00293-t004:** Summary of studies: Autonomic Nervous System and Irritable Bowel Syndrome (IBS).

Sr.	Study	Population (n)	Study Design	HRV Parameters	Results	Conclusion
1	Pellissier et al., 2010 [73]	Patients: IBS = 27 (9M, mean age = 40 ± 14 yrs) Healthy subjects = 21 (8M, mean age = 39 ± 12 yrs)	HRV was recorded after 30 min of relaxation period for 10-min.	HR, lFnu, hFnu, LF/HF, VLFnu	The values of lFnu and LF/HF were significantly higher while those of hFnu was significantly lower in IBS patients in comparison to that of healthy controls.	IBS was associated with a high sympathetic tone and low parasympathetic tone.
2	Kano et al., 2019 [16]	Patients with non-constipated IBS = 27 Healthy Controls = 33	HRV was assessed for 5-min during baseline, 3-min resting with bag in the rectum and 3-min during tonic distension. Brain responses to colorectal distention were measured using fMRI and correlated with individual HRV parameters.	*ln*(HF power), *ln*(LF/HF), *ln*(HR)	*ln*(LF/HF) was low in the IBS group in response to colorectal distention compared with controls (*p* = 0.003). The baseline *ln*(HF) was negatively correlated with toleration threshold to the rectal distention in controls only.	The IBS patients exhibited abnormal interactions between ANS activity and the brain mechanisms underlying visceral perception.
3	Tillisch et al., 2005 [74]	130 IBS patients 55 healthy controls Subgroups for sigmoid balloon distension study: 46 IBS patients 16 healthy controls	For ANS assessment, ECG was recorded for 10 min at the end of the 20-min initial rest period in all subjects. In subgroup analysis, ECG was recorded during last 10 min of a 20-min rest period following sigmoidoscopic balloon placement, and during the 10-min period of phasic sigmoid distensions	Peak power ratio (LF/HF), peak power high frequency (PPHF)	IBS patients had higher PPR and lower PPHF across conditions. Male IBS patients had higher PPR and lower PPHF as compared to female patients.	IBS patients showed increased sympathetic and decreased parasympathetic activity compared to healthy controls. These variances were mainly observed in males.
4	Robert et al, 2004 [75]	70 women with IBS divided into 2 groups, with and without depressive symptoms: IBD-DS (n = 44, age = 33 ± 1.1 yrs) IBS + DS (n = 26, age = 40.1 ± 2.6 yrs) Healthy controls (n = 21, age = 34.7 ± 1.9 yrs)	HRV analysis was carried out for 15-min segments selected from 1-baseline pre-sleep period 2-stage 2 sleep 3-slow-wave sleep 4-rapid-eye movement sleep	LF/HF ratio	No significant difference was observed in LF/HF values for any group compared to HC during any type of sleep.	No difference in LF/HF at baseline or in any sleep stages between IBS patients with/without depressive symptoms and health controls.
5	Jarrett et al., 2015 [76]	In a two-armed randomized controlled trial Comprehensive self-management (CSM) group (n = 41, age = 40.5 ± 14.6 yr) was compared to a usual care (n = 44, age = 37.9 ± 15.9 yr) group.	ECG was recorded for each participant 3 and 6 months after randomized controlled trials for (1) CSM and (2) usual care groups to test whether these HRV predicts improvements in primary outcome, including abdominal pain, GI symptom score, and IBS-specific quality of life.	HF power, LF/HF ratio	Participants with lower nighttime HF power and higher LF/HF had less benefit from CSM on abdominal pain.	Patients with higher sympathetic tone were less benefitted from the cognitively focused therapies to reduce the IBS related abdominal pain.
6	Jang et al., 2017 [15]	43 participants (age = 21.4 ± 2.1 yr) with IBS-C Subgroups: cognitive behavior therapy (CBT) group (n = 23, age = 21.6 ± 1.8 yr) received cognitive behavior therapy and control group (n = 20, age = 21.2 ± 2.4 yr)	All the participants completed a questionnaire assessing their GI symptoms and their HRV was measured via ECG at baseline and 8, 16, and 24 weeks.	HF power, LF/HF ratio	In CBT group, the HF power was higher and LF/HF ratio was lower than that of control group at 8 weeks. Changes in the HF power were significantly and inversely associated while changes in LF/HF ratio were significantly and positively associated with changes in GI symptoms, anxiety, depression, and stress at 16 and 24 weeks.	ANS measurement via HRV could be a valuable objective parameter for evaluating response to CBT in young IBS-C patients.
7	Tanaka, Kanazawa, Palsson et al., 2018 [17]	156 Rome III positive IBS patients (131F, mean age = 35 ± 11 yr) 31 healthy controls (24F, mean age = 37 ± 13 yr)	All the participants underwent colonic manometry with descending colon bag distention, followed by an 810-kcal meal. HRV was measured during baseline, colonic distention, and after the meal.	HF power (%), LF power (%), LF/HF ratio	%HF was decreased, and the LF/HF ratio was increased by both eating and colonic distension. In IBS patients, the %HF and LF/HF ratios were correlated with psychological symptoms but not bowel symptoms.	In IBS patients, HF and LF/HF were correlated with psychological symptoms but not bowel symptoms.
8	Cheng et al., 2013 [77]	Rome III positive IBS patients (n = 35, 53% F, mean age = 37.89 yrs) and healthy controls (n = 31, 58% F, mean age = 37.26 yrs)	All of the participants underwent ECG recording and plasma catecholamines measurement at rest and during flexible sigmoidoscopy (FS).	HF power, LF/HF ratio	At rest, IBS patients had slightly lower HF and LF/HF compared to controls (not statistically significant). During FS, controls showed a transient increase in LF/HF and decrease in HF power. IBS patients had significantly less sympathetic and vagal responsiveness both before and after the stimulus. Those with longer period of disease had less sympathetic and vagal responsiveness than those with shorter period.	Dysregulated ANS responses to a visceral stressor were observed in IBS patients, which might be associated to the duration of the disease.
9	Jarrett et al., 2012 [78]	Children with functional abdominal pain (FAP) or IBS (n = 100, 70F, age = 8.9 ± 1.1) Healthy controls (n = 62, 44F, age = 9.3 ± 1.1)	Participants completed a questionnaire, filled a 2-week pain/stool diary, and wore a 24-h Holter monitor to assess vagal activity. HRV was compared for both groups.	HF power, LF/HF ratio	No difference was observed in the vagal HRV parameters among FAP/IBS children and HC. A negative correlation between vagal activity and psychological distress was observed in FAP/IBS girls but not boys.	In girls with FAP/IBS, there was an inverse relationship between vagal activity and psychological distress.
10	Davydov et al., 2016 [79]	78 female IBS patients (age = 35.3 ± 12.8 yrs) 27 healthy females (age = 33.3 ± 11.7 yrs)	All of the participants were assessed for IBS symptoms, blood pressure (BP), HR, HRV, and baroreceptor sensitivity (BRS) at rest.	*Ln*(HF), *ln*(LF)	Lower BRS was associated with higher IBS severity if the effect was transferred through the decrease of LF power while it was associated with lower IBS severity if the effect was transferred through diastolic BP increase. Lower BRS was associated with higher abdominal pain severity if the effect was transferred through the decrease of HF power.	The IBS development and its symptoms’ severity were associated to different cardiovascular mechanisms.
11	Jarrett et al., 2016 [80]	Patients with IBS (n = 47) Healthy Controls (n = 29) All women with age between 18 and 49 years. Patients subgroup: IBS-diarrhea [IBS-D] (n = 22) IBS-constipation plus mixed [IBS-CM] (n = 25)	All of the Participants kept a four-week symptom diary and had a 12-h Holter placed to assess night-time HRV	HF power, LF power, TP, LF/HF, HR	No HRV parameter differ between the groups or IBS subgroups.	Spectral HRV parameters could not differentiate the IBS-D, IBS-CM, and/or healthy controls.
12	Walker et al., 2017 [81]	Pain-remit patients (n = 130, 58.5%F, age = 20.50 ± 3.33 yrs) Pain-persist patients (n = 96, 71.6%F, age = 19.73 ± 3.68 yrs) Healthy controls (n = 123, 53.7%F, age = 18.25 ± 2.72 yrs)	Recorded ECG data at rest and during laboratory stressors.	SDRRI, HF power	Pain-persist females had significantly lowered SDRRI, and HF power compared to males in their group, as well as to that of female controls, and all males regardless of pain category.	Young women with persistent functional abdominal pain showed a low parasympathetic and high sympathetic tone.
13	Pellissier et al., 2014 [82]	Healthy controls (n = 26, 18F, age = 36 ± 10 yrs) Patients with IBS (n = 26, 19F, age = 38 ± 10 yrs) Patients with Crohn’s disease (CD) (n = 21, 12F, age = 40 ± 11 yrs)	After 30 min of rest, ECG was recorded for HRV analysis. Blood samples were taken after the ECG recording for plasma cortisol, epinephrine, norepinephrine, TNF-alpha, and IL-6 levels.	hFnu	Control subjects with higher hFnu had significantly lower evening salivary cortisol levels. This was not observed in CD and IBS patients. Negative correlation was observed between the hFnu and TNF-alpha level in CD patients. In IBS patients, vagal tone was inversely correlated with plasma epinephrine.	HF was negatively correlated with TNF-alpha in CD patients and negatively correlated with plasma epinephrine in IBS patients.
14	Tanaka, Kanazawa, Kano et al., 2018 [83]	32 patients with IBS (16F, age = 21.7 ± 1.6 yrs) 32 healthy controls (16F age = 22.0 ± 2.1 yrs)	The patients received no, mild, or strong colorectal distension. The heart rate and HRV were analyzed using ECG. At each distension, plasma noradrenaline, adrenaline, adrenocorticotropic hormone (ACTH), and cortisol levels were taken.	LF power, HF power, LF/HF ratio	In controls a strong correlation was observed between adrenaline and HRV upon corticotropin-releasing hormone (CRH) injection, but not IBS patients.	The relationship between HPA-sympathoadrenal responses and CRH levels during colorectal distention differed between patients with IBS and controls.
15	Karling et al., 1998 [84]	18 IBS patients (4 men, mean age = 20.6 ± 49.2 yrs) 36 healthy controls (mean age = 31.4 yrs)	After 10 min rest, ECG blood pressure and respiration rate were recorded in supine position in normal breathing followed by 12 br/min breathing rate followed by 70° tilt and for 3-min after the tilt. HRV parameters were compared between IBS patients and controls	LF power (called mid frequency in this paper) HF band	LF power in IBS patients was significantly higher compared to the healthy controls during supine and tilt. No significant difference in HF power was observed.	IBS patients had significantly increased sympathetic activity while parasympathetic activity was same as controls.
16	Cain et al., 2007 [85]	Healthy controls (n = 50) Women with IBS (n = 165) *Subgroups:* Constipation predominant (n = 45); diarrhea predominant (n = 64) alternating (n = 56)	24 h ECG was recorded to assess autonomic activity for all the participants and the HRV parameters’ values were compared among subgroups and against control. The severity of the gut pain was also recorded.	LF power, HF power, LF/HF ratio	No difference in HRV parameters was found upon comparing 165 IBS patients and 50 healthy controls. The women with severe gut pain had lower value HF power and higher value of LF/HF ratio compared to the women with no severe gut pain among all groups.	IBS patients with severe gut pain have lower parasympathetic tone and predominant sympathetic activity compared to those with no severe gut pain.
17	Polster et al., 2018 [86]	Healthy controls (n = 39, age = 29 yrs, age range = 19–49, BMI = 22.8 ± 3.3) IBS patients (n = 158, age = 35, age range = 19–64, BMI = 23.4 ± 3.9)	HRV was measured in supine, standing with controlled respiration, and ambulatory 24 h period. Frequency domain parameters (5 min, supine and standing) and time domain parameters (24 h) were calculated and compared between patients and controls.	SDNN, RMSSD, pNN50%, SDNN index, lFnu, hFnu, LF/HF ratio	During the daytime, all the time domain parameters (SDNN, RMSSD, pNN50, SDNN index) were significantly lower in patients compared to controls. No difference was observed during nighttime. HF power, LF power, and LF/HF ratio was significantly different only during standing. No difference was observed during supine. A subgroup of patients with aberrant HRV values had more severe symptoms.	Patients with IBS had altered ANS function compared to controls.
18	Orr et al., 2000 [87]	15 IBS patients (13f, mean age = 34.9 ± 2.1 yrs) 15 healthy controls (13 f, mean age = 36.2 ± 2.3 yrs)	ECG was recorded during 1 h pre-sleep walking and both ECG polysomnography was recorded during 7 h sleep.	HF power, LF power, LF/HF ratio	LF power value was significantly higher during walking and LF/HF ratio was significantly higher during rapid eye movement sleep in IBS patients. No difference in HF power was observed.	IBS patients had higher sympathetic activity during walking and overall sympathetic dominance during rapid eye sleep.
19	Adeyemi et al., 1999 [88]	35 IBS patients (mean age = 39.1 ± 9.5 yrs, M:F ratio 2.9:1) and 18 healthy controls (mean age = 38.2 ± 6.5 yrs, M:F ratio 2:1)	EKG signal was recorded during supine, standing, and deep breathing.	VLF power, LF power, and HF power	IBS patients had significantly high VLF power in supine, high HF power during standing, and low HF power during deep breathing.	IBS patients had reduced sympathetic influence in response to standing and reduced PNS modulation during deep breathing.

**Table 5 diagnostics-13-00293-t005:** Summary of studies: Autonomic Nervous System and Constipation.

Sr.	Study	Population (n)	Study Design	HRV Parameters	Results	Conclusion
1	Liu et al., 2022 [38]	14 patients with chronic constipation 21 healthy controls	All the participants underwent active standing test and high-resolution colonic manometry (HRCM) with concurrent ECG recording. The autonomic reactivity to the postural change and to the stimuluses during HRCM (meal, balloon distensions, rectal bisacodyl) of the patients were compared to those of healthy controls.	SI, RSA, RMSSD, SI/RSA	In response to standing, SI/RSA was significantly higher in patients compared to healthy controls. 12 of the 14 patients had high (SI) and/or low PNS (RSA, RMSSD) reactivity to either vagal or sacral stimuli. Both patients and controls had similar baseline values of HRV parameters.	High sympathetic tone and reactivity was more responsible for refractory chronic constipation as compared to vagal impairment.
2	Ali et al., 2022 [24]	41 patients (28F, age = 37 ± 17 yrs). Chronic constipation (n = 28), fecal incontinence (n = 5); both constipation and fecal incontinence (n = 8)	All of the participants had lumbar and sacral neuromodulation via low level laser stimulations (LLLS). ECG was recorded for baseline, LED array stimulations, laser probe stimulations and recovery.	RMSSD, RSA, SI, SI/RMSSD, SI/RSA	Laser probe stimulations increased RSA, RMSSD, and decreased SI, SI/RSA, and SI/RMSSD values compared to baseline	Neuromodulation of lumbar and sacral autonomic nerves of patients with constipation via LLLS increased the PNS and decreased the SNS activity.
3	Wang et al., 2019 [89]	*Animal Study:* 10 adult male Sprague-Dawley (SD) rats, weighting 250–300 g	The rats were given loperamide (Lop) to induce constipation and electroacupuncture (EA) was applied via a pair of electrodes implanted at ST-36. The effects of actual and sham EA on were evaluated via HRV.	LF power, HF power, LF/HF ratio	The HF power was increased and LF power was decreased by EA in both normal rats and rats treated with Lop. The LF/HF ratio was reduced by EA in both normal and Lop-treated rats.	Rats with loperamide-induced constipation exhibited decreased vagal and increased sympathetic activities, which could be reversed by EA at ST36.
4	Miyagi et al., 2022 [90]	Patients with Parkinson’s disease (n = 17, mean age = 66 yrs, age range = 60–72, 47%F, 76% constipated) Healthy controls (n = 11, mean age = 63 yrs, age range = 59–64, 55%F, 9% constipated)	Subjects underwent ECG recording during supine and standing. Coefficient of variations of RR intervals (CVRR) was calculated during rest and deep breathing. The values were compared between control group and PD group with constipation as a dominant symptom.	LF power, HF power, LF/HF ratio	During both supine and standing, the patients had low HF and LF power values compared to those in controls. However, the LF/HF ratio was higher during supine and lower during standing in patients. All of the differences were not statistically significant. The resting LF power was associated with constipation	Sympathetic and parasympathetic nerves were concurrently damaged in patients with PD.
5	Huang et al., 2019 [91]	*Animal Study:* Thirty Sprague-Dawley rats (male, 300–350 g). *Subgroups:* SNS group (n = 10); Sham-SNS (n = 10); control Rats (n = 10)	The effects of SNS with optimized parameters on loperamide induced constipation were studied. SNS was performed at S3 sacral nerve unilaterally (right side of the body), 4 h daily for 7 days in rats and ECG was recorded for autonomic functions assessment. HRV parameters were compared between SNS, Sham-SNS and control groups.	HF power, LF power, LF/HF ratio	Sacral nerve stimulation significantly increased HF power and decreased both the LF power and LF/HF ratio as compared to both the Sham-SNS and control groups.	Sacral nerve stimulation with optimized parameters improved constipation by restoring impaired ANS in rats with loperamide-induced constipation.
6	Z. Liu et al., 2018 [92]	Patients with ischemic stroke divided into two groups. Sham-TEA (n = 44, 30M, age = 68.3 ± 12.3 yrs) TEA (n = 42, 27M, age = 70.1 ± 11.2 yrs)	One group received transcutaneous electrical acustimulation (TEA) while another group received Sham-TEA for 2 weeks. ECG was recorded for the assessment of ANS. Constipation and dyspeptic symptom assessment was performed at the end of the 14-day treatment.	HF power, LF power, LF/HF ratio	TEA significantly increased the value of HF power and decreased LF power as well as LF/HF ratio significantly while Sham-TEA did not.	TEA could be used to mediate the autonomic function to treat the stroke-induced constipation
7	Jin et al., 2015 [93]	*Animal Study:* Five adult female hound dogs (3–4 yr, 22–25 kg)	Impaired colonic motility was induced by rectal distension (RD) in dogs. Colon contractions and transit were measured in various sessions with and without electroacupuncture (EA) and ECG was recorded for autonomic assessment.	HF power, LF power, LF/HF ratio	The decrease in HF power and increase in sympathetic activity due to rectal distension was significantly normalized (reversed) by EA.	EA at ST36 reinstated the RD-induced impairment in both colonic contraction and transit by increasing vagal activity
8	Gondim et al., 2021 [94]	41 patients with overactive bladder (19 out of 41 constipated) 20 healthy controls aged between 5 and 17 years	HRV was recorded for 1 min during empty bladder (pre-voiding), the full bladder, and 5 min after the spontaneous voiding (post-voiding) into uroflowmeter.	Mean RR, SDNN, pNN50%, lFnu, LF power, hFnu, HF power, LF/HF ratio	Higher heart rate variability was observed in control group. LF was predominant in the control group only during post-voiding.	Constipation was associated with increased sympathetic tone.
9	J. Liu et al., 2020 [95]	45 patients with functional outlet obstruction constipation (FOOC) *Subgroups:* Group A (n = 15, 11F, age = 47.20 ± 10.98 yrs) Group B (n = 15, 11F, age = 45.38 ± 12.79 yrs) Group C (n = 15, 12F, age = 43.24 ± 9.22 yrs)	Group A received Macrogol 4000 Powder (MAC) only twice a day, group B received adaptive biofeedback (ABT) +MAC+ Sham-TEA twice a day while group C received TEA + ABT (30-min) + MAC twice a day for 6 weeks study. HRV was calculated at baseline and after each corresponding therapy	HF/(HF + LF), LF/(HF + LF)	LF/(LF + HF) in groups B and C were decreased and HF/(LF + HF) was increased in groups B and C as compared to group A.	Both sham and actual TEA modulated the ANS in patients with FOOC.
10	Enevoldsen et al., 2018 [96]	Patients with acquired brain injury (ABI) (n = 25, 18M, age = 61.3 [30.7–74.5] yrs); controls (n = 25, 18M, age = 61.5 [34–70] yrs)	Gastrointestinal transit time (GITT) was measured using radio-opaque markers. HRV was recorded from 24 h and 5-min analysis of ECG recorded using 1-lead wearable device. HRV parameters were correlated with GITT. GITT was also compared between patients and controls.	SDNN, RMSSD, LF power, HF power, LF/HF, total power (TP)	No correlation was observed between GITT and any of the HRV parameters.	Patients with ABI showed prolonged GITT compared to that of controls, which was not related to HRV.
11	J. Liu et al., 2022 [97]	Patients with functional constipation (FC): *Subgroups:* With sleep deficiency (SD) (n = 85, 38M, age = 47.39 ± 9.58 yrs). Without sleep deficiency (n = 193, 76M, age = 46.64 ± 10.09 yrs);	All the participants underwent high-resolution anorectal manometry, they were required to fill in a questionnaire about constipation symptoms. ECG was recorded to generate the HRV parameter for the autonomic function comparison of two groups.	SI, HF/(LF + HF)	SD is associated with higher score of constipation. Constipated patients with SD had significantly low HF/(HF + LF) and significantly higher SI compared to those without SD.	Constipation was associated with sleep deficiency and impaired autonomic function.
12	N. Zhang et al., 2014 [98]	12 patients with chronic functional constipation with at least 1 year history of symptoms	All of the patients underwent 2-weeks of transcutaneous neuromodulation (TN) and 2-weeks of sham-TN with 1 week washout in between. TN and sham-TN were applied at posterior tibial nerve and the acupoint ST36. The ECG was recorded along with symptoms, quality of life score, and anorectal motility.	HF power, LF power, LF/HF ratio	HF power was increased and both LF power and LF/HF ratio decreased during TN compared to the baseline.	TN at posterior tibial and ST36 was effective in chronic constipation; the effect was possibly mediated via autonomic nervous system mechanism.
13	Ding et al., 2012 [99]	21 patients with functional constipation (mean age 44.8 ± 16.2 years) Healthy controls (n = 115; females = 70; average age = 50 yrs)	The psychological status, quality of life and autonomic nervous system activity was recorded using heart rate variability with HANS-1000 autonomic biofeedback apparatus, before and after 10 sessions of biofeedback training.	HF power, LF power, LF/HF ratio	After 10 sessions of biofeedback, the HRV parameters did not change statistically although a slight change in absolute values of HF and LF power was observed.	It was not established that biofeedback training improves the autonomic nervous system in patients with chronic constipation.
14	Wu et al., 2020 [100]	18 patients (M/F: 9/9) with functional constipation	A cross-over study was performed with 2-weeks TN at acupoint ST36 and a 2-week TN at (posterior tibial nerve) PTN. Change in constipation symptoms, QOL anorectal manometry and autonomic function was recorded before and after each case.	HF power, LF power, LF/HF ratio	HF power was increased significantly due to TN at both the points (ST36 and PTN). While the other parameters, such as symptoms, QOL score, number of weekly bowel movements were improved more in case of TN at ST36.	TN at both ST36 and PTN increased the vagal activity of the patients with functional constipation.
15	C.-Y Chen et al., 2010 [101]	36 patients (ave age = 40 ± 12 yrs) with constipation.	The participants were divided into two groups: electro-acupuncture (EA) group and sham (SA) group. The EA group patients were pricked respectively at ST36, ST37, ST25, ST28, CV4, and CV6 for 8 weeks. The HRV parameters were recorded at 1st, 4th, and 8th weeks.	hFnu, lFnu, LF/HF ratio	Both lFnu and LF/HF ratio decreased significantly after 8 weeks in EA group only. hFnu was significantly higher than SA group after 8 weeks; however, it was not different than the same group at week 1.	Electro-acupuncture decreased the sympathetic nervous system with 8 weeks of treatment.

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
