# Peer review of "Roles of Heart Rate Variability in Assessing Autonomic Nervous System in Functional Gastrointestinal Disorders: A Systematic Review"

_diagnostics, 2023, doi:10.3390/diagnostics13020293_

Round 1
Reviewer 1 Report
Functional GI disorders (FGID) are as the authors clearly state and provide data, are common worldwide and use significant medical resources. Unfortunately, they are not always investigated to the extent of other disorders and studies on specifically the role of neural dysfunction such as enteric or autonomic/enteric dysregulation are uncommon. This is true although as the authors state, some research has been done on brain regulation of gut functions. The hypothesis I believe is to utilize changes in cardiac function which are likely due to autonomic dysregulation to correlate with FGID, a novel and thought provoking hypothesis. It may not be and likely is not that the cardiac dysregulation mediates or causes the FGID, this if true should be clearly stated. Thus one item that would enhance the manuscript is could the authors add a section on other ways to determine autonomic effects on GI functions in addition to the changes in the cardiac parameters? This short section should include a consideration of standard measures of autonomic functions, those physiological functions of various organs regulated by the autonomic nervous system. Some may not be correct and appropriate as they may not relate well to changes in autonomic regulation of enteric nervous system regulation of GI functions and this should be stated. Also the validity of using changes in cardiac function to assess how autonomic changes regulate GI functions should be presented in greater detail. Establishing the validity has been presented, but should be expanded.
It should be stated that the methodology used is well designed and clearly presented. The data presentation in the tables such as tables 2-5 is pivotal for the clarity of communication what the authors wish to convey and are very good. However in the first few paragraphs I would strongly encourage clearer presentation of the hypothesis and its implications.
Reviewer 2 Report
This is an interesting review of the literature related to assessment of the HRV-based assessment of autonomic function to identify autonomic dysfunction in selected FGIDs. The review of the literature is a bit limited in that only PubMed was searched and only free full text articles were selected which may mean high-quality, peer-reviewed articles may have been omitted from the review.
One of the FGIDs included was gastroesophageal reflux (GERD) which is not included as an FGID in the Rome IV criteria. It is not clear why this is considered a functional disease in this paper. Please clarify the selection of entities.
There is good background information though some of it would benefit from references such as the information related to FGIDs arise mainly due to impaired central nervous system process of sensory input. The references are minimal, and the background information would be stronger if adequately referenced.
It does not appear that the PRISMA criteria for reviews was used. More information is needed related to the choice of articles for the review and what is meant by short-listed. The selection process needs more detail to include the methods used to decide whether a study met the inclusion criteria of the review, including how many reviewers screened each record and each report retrieved, whether they worked independently, and if applicable, details of automation tools used in the process. The methods of data collection also need to be specified, including how many reviewers collected data from each report, whether they worked independently, any processes for obtaining or confirming data from study investigators, and if applicable, details of automation tools used in the process.
The tables appear fairly complete. The review of the articles is limited to basic description of the various components, but further synthesis is needed. References need to be included as studies are noted. As noted in the PRISMA criteria, describe any methods used to explore possible causes of heterogeneity among study results (e.g. subgroup analysis, meta-regression) and any sensitivity analyses conducted to assess robustness of the synthesized results.
The discussion is a reasonable review but limited. Again, references are needed for a number of the statements. Limitations of the review are not noted and needed to be included.
It would be good to register the review and protocol per PRISMA criteria.
Round 2
Reviewer 2 Report
The authors have addressed most of the concerns. There is still the concern of using only PubMed and only full texts which may result in missing articles. The methods are better described, however there is review by only one author so credibility comes into question. Results should confirmed by another author.
The other concern is that GERD is called a FGID which is not consistent with the ROME criteria or general GI literature. While it can included in the review, it needs to clear that it is not a FGID.
Author Response
Comment: The authors have addressed most of the concerns. There is still the concern of using only PubMed and only full texts which may result in missing articles.
Response: We have now researched the articles using "web of science", Elsevier/Science Direct, and Scopus in addition to PubMed and included all the retrieved articles in our pool of studies. We have added 16 new papers to our review (Tables 2-5).
Comment:The methods are better described, however there is review by only one author so credibility comes into question. Results should confirmed by another author.
Response: All the search criteria and results are reconfirmed by other author as well.
Comment: The other concern is that GERD is called a FGID which is not consistent with the ROME criteria or general GI literature. While it can included in the review, it needs to clear that it is not a FGID.
Response: we have clarified in the paper about GERD being not am FGID